# How to Predict the Innovation to SMEs? Applying the Data Mining Process to the Spinner Innovation Model

Ronnie Figueiredo [1,2,3,*], Carla Magalhães [4] and Claudia Huber [4]

1    Research Center in Business Sciences, NECE (UBI), 6200-209 Covilhã, Portugal
2    Centre of Applied Research in Management and Economics (CARME), School of Technology and Management (ESTG), Polytechnic of Leiria, 2411-901 Leiria, Portugal
3    Spinner Innovation Centre (SIC), 2840-626 Setúbal, Portugal
4    TRIE—Transdisciplinary Research Center for Entrepreneurship & Innovation Ecosystems, Lusófona University, 4000-098 Porto, Portugal
*    Correspondence: ronnie.figueiredo@ipleiria.pt

**Abstract:** Despite the importance of small and medium-sized enterprises (SMEs) for the growth and development of companies, the high failure rate of these companies persists, and this correspondingly demands the attention of managers. Thus, to boost the company success rate, we may deploy certain approaches, for example predictive models, specifically for the SME innovation. This study aims to examine the variables that positively shape and contribute towards innovation of SMEs. Based on the Spinner innovation model, we explore how to predict the innovation of SMEs by applying the variables, namely knowledge creation, knowledge transfer, public knowledge management, private knowledge management and innovation. This study applied the data mining technique according to the cross industry standard process for data mining (CRISP-DM) method while the Statistical Package for the Social Sciences (SPSS_Version28) served to analyze the data collected from 208 SME employees in Oporto, Portugal. The results demonstrate how the Spinner innovation model positively influences the contributions of the SMEs. This SME-dedicated model fosters the creation of knowledge between internal and external interactions and increases the capacity to predict the SME innovation by 56%.

**Keywords:** Spinner innovation; data mining; predictive; model; open innovation

## 1. Introduction

In the 1990s, when intellectual capital began to gain greater prominence, corporate knowledge management took its place in the corporate agenda (Davenport and Prusak 2003). Thus, knowledge has turned into a very valuable competitive resource in keeping with its capacity to nurture innovation and competitive sustainability (Davenport and Prusak 2003).

Furthermore, the works of Nonaka, in 1991, and Nonaka and Takeuchi, in 1995, greatly contributed to this close link between innovation and knowledge creation. Indeed, there has since arisen the assumption that innovation inherently involves the generation of new knowledge. Several authors shared this point of view, including Subramaniam and Youndt (2005) who describe how innovation consists of continuously striving to apply new and unique knowledge. Similarly, Du Plessis (2007) interlinks innovation with the creation of new knowledge and ideas to facilitate new business results and, in turn, Lundvall and Nielsen (2007) maintain that innovation represents something new and, therefore, adds an additional facet to the existing knowledge. In addition, Stewart (1997) argues that an organization's ability to innovate depends considerably on the knowledge of its teams, the knowledge incorporated into business processes and customer relationships.

We may thus verify that knowledge has in recent years gained widespread recognition as one of the most crucial competitive organizational assets (Palacios-Marqués and Garrigos-Simon 2006). In fact, knowledge management has become a very common issue in the 21st

century in keeping with its application in a wide spectrum of activities and areas with the objective of managing, creating and enhancing intellectual assets (Shannak 2009).

Furthermore, it is important to note that in the current complex and dynamic environment, for organizations to be able to foster competitive advantage and optimize their organizational performance, they must create and transfer new knowledge and practices. Based on this perspective, Ichijo and Nonaka (2007) state that the success of any organization in the 21st century will reflect its ability to develop intellectual skills through the creation and transfer of knowledge (Webster 2008).

This question leads onto another reflection around the ways in which innovation and knowledge intersect, which establishes a distinction between open and closed innovation (Panagopoulos 2016). Open (public) innovation incorporates the new trends in industry, with startups and spinoffs increasingly relevant, and the development of innovations are often shared. In contrast to open innovation, closed (private) innovation conveys the paradigm that prevailed for most of the 20th century. However, although open innovation has received more attention in the last decade (Hippel and Krogh 2003; Laursen and Salter 2006), there are studies that report how certain forms of governance best address particular types of innovation problems, whether open or closed (Felin and Zenger 2014).

A study undertaken by Almeida (2021) to explore the diversity of open innovation practices adopted by Portuguese SMEs, took into consideration the outside-in, inside-out and coupled paradigms. The study concluded that organizations favoured the adoption of the outside-in paradigm with the most commonly adopted outside-in practices being the integration of external knowledge from suppliers and customers. The most relevant benefit reported was the enhanced innovation capacities of these organizations. This study holds particular relevance for establishing state support policies capable of enabling the involvement of Portuguese SMEs in open innovation processes.

Remaining with the Portuguese SME context, there is every importance in analyzing the issue of innovation forecasting. We may correspondingly assume that SMEs can develop and renew their ability to manage and transfer knowledge, actively implementing the acquisition of external knowledge and transferring this knowledge internally. Furthermore, these capacities to manage and transfer knowledge contribute both to company innovation and their performance (Zhou and Uhlaner 2009).

Indeed, the relevance of applying these concepts to SMEs also extends to their positioning as organizational actors, especially vulnera\ble to market fluctuations, as recently demonstrated by the COVID pandemic.

Therefore, this study seeks to approach the theme of predicting innovation in SMEs in accordance with the Spinner innovation model, in accordance with the following departure question: *How to predict the innovation of SMEs?*

The subsequent article structure is as follows: Section 2 presents a literature review on the theme approached by this study; Section 3 defines the research methodology and presents both the hypothesis and the results; Section 4 sets out the discussion and study implications; and Section 5 details the conclusions, limitations and future research directions.

## 2. Literature Review

### 2.1. Predictive Models

Predictive models provide inputs to many real-life scenarios (Hyndman and Athanasopoulos 2018). They often establish the base grounds for many decision-making procedures (Shim 2000), including health-related issues, human resource requirements (O'Brien-Pallas et al. 2001) and expenditure calculations (Lee and Miller 2002), for example. Furthermore, in an era of global competition, innovation has become a central objective to achieve sustainable futures for organizations capable of coping with the challenges faced.

Despite the importance of SMEs to the growth and development of companies, these companies continue to return high failure rates and this demands attention from the management community. Thus, deploying certain approaches can help boost the success rate of entrepreneurs. Some studies have already verified the validity of predictive models,

with some cases specifically including SMEs. One such study analysing entrepreneur demographics as a predictor of success among SMEs focused on the case of Lagos State, Nigeria (Genty et al. 2015). Another example involves the application of corporate failure diagnosis models to SMEs with the aim of not only identifying those predictor variables capable of raising the accuracy of the diagnoses generated by corporate failure models, but also carrying out comparative analyses of the proposed models in the existing literature, focusing on the case of SMEs in Greece (Kosmidis and Stavropoulos 2014), for example. Another study seeks to ascertain the impact of transformational leadership (TL), knowledge management (KM), citizenship behaviour or positive deviance (PD) and intrinsic motivation (IM) on organizational innovation (OI). One case study considers the situation of SME ready-made garment manufacturers in Bangladesh (Fan et al. 2017).

### 2.2. Knowledge Management

Knowledge management (KM) emerges as a critical factor for sustainability in the increasingly competitive and evolving manufacturing industries (Langley et al. 2017). The literature correspondingly often defines KM and innovation as the key drivers for improving organizational performance (Ngoc-Tan and Gregar 2018).

Drucker (1993) describes knowledge as the only meaningful resource in a knowledge society. He further stresses that: "Knowledge is not impersonal like money. Knowledge does not reside in a book, a data bank, or a software program. They contain only information. Knowledge is always embodied in a person, taught and learned by a person, used or misused by a person" (Drucker 1993, p. 191).

The main objective of KM practices is to maximize individual knowledge by extracting tacit and implicit knowledge and translating it into explicit knowledge, which then becomes subject to interpretation, representation, codification, storage, retrieval, sharing and dissemination (Nunes et al. 2006).

In general, KM represents a strategy for managing organizational knowledge assets in support of management decision-making, increased competitiveness and innovation and creativity capabilities (Zyngier et al. 2004). In operational terms, KM embodies a cycle that begins with knowledge creation and then is followed by its interpretation, dissemination and retention (De Jarnett 1996).

However, there is no single approach to KM, with some authors differentiating between technical types and strategic types (Liebeskind 1996). Grant (1996) proposes practical knowledge, intellectual knowledge (scientific, humanistic and cultural), hobby knowledge (news, gossip and stories) and unwanted knowledge.

Furthermore, the most common characterization of knowledge spans tacit, explicit and implicit Knowledge (Nonaka 1994; Nonaka and Konno 1998).

Although KM has undergone successful application in large companies, SMEs also need adequate and up-to-date knowledge to compete, especially because SMEs tend to be more vulnerable to problems including high staff turnover and knowledge retention. Consequently, companies must appropriately manage, disseminate and retain this knowledge. Although KM processes may incur direct and indirect costs, the consequences of SMEs not maintaining these processes potentially involve knowledge leakage and consequent losses in efficiency, productivity, competitiveness and greater SME vulnerability (Nunes et al. 2006).

### 2.3. Knowledge Creation and Transfer

The creation and transfer of knowledge in organizations have become critical factors for their success and competitiveness.

Knowledge creation deploys a continuous process through which one overcomes individual boundaries and the constraints imposed by information and past learning by acquiring new contexts, new worldviews and new Knowledge (Saenz et al. 2009). By interacting and sharing tacit and explicit knowledge with others, individuals enhance their

capacities to define a situation or problem and apply their respective knowledge to act and specifically solve the problem (Nonaka et al. 2006).

In the case of organizational knowledge creation, this means making available and amplifying the knowledge created by individuals, as well as crystallizing and connecting it with the organization's knowledge system (Nonaka and Takeuchi 1995; Nonaka et al. 2006). Knowledge creation refers to the ability of an organization to develop novel and useful ideas and solutions (Marakas 1999). By reconfiguring and recombining foreground and background knowledge through different sets of interactions, organizations are able to create and design new realities and meanings (Bhatt 2001).

Knowledge transfer involves two actions: transmission (sending or presenting the knowledge to a potential recipient) and absorption by that person or group. The literature further stresses how transmission and absorption hold no inherent value unless they lead to changes in behavior and/or to the development of ideas that lead to new behaviors (Davenport and Prusak 1998).

Hansen (1999) suggested two strategies for the transfer of organizational knowledge: "codification" and "personalization". Through codification, all knowledge becomes standardized, structured and stored in information systems. In contrast, personalization emphasises tacit knowledge transfers from one person to another (Brachos et al. 2007).

The organizational context determines the ways in which knowledge undergoes creation, legitimation and diffusion throughout the organization (Davenport and Prusak 1998). Furthermore, the notions of organizational context, culture and social climate represent overlapping perspectives on the same phenomena (Ashkanasy et al. 2000; Smith et al. 2005).

The factors identified as influencing and characterizing favorable contexts for knowledge transfers include social interaction, trust, management support, motivation and learning orientation (Brachos et al. 2007). According to Nonaka and Takeuchi (1995) knowledge sharing represents a critical stage in knowledge transfer processes.

Therefore, knowledge creation and transfer are both essential factors for creating new knowledge and producing innovation (Dalkir 2005).

### 2.4. Innovation

O'Sullivan and Dooley (2009, p. 1) define innovation as "the process of making changes to something established by introducing something new that adds value to customers".

The literature features many studies on the determinants of organizational innovation. Damanpour and Schneider (2006) understand that the determinants of innovation span three different levels of analysis: managerial, organizational and environmental. In turn, Crossan and Apaydin (2010) propose a multidimensional structure of organizational innovation that includes innovation leadership, managerial levers and business processes as well as dimensions pertaining to innovation as a process alongside other dimensions relating to innovation.

The literature also describes the innovation process as segmented into phases: the frontend phase (idea generation and concept development), the product/service development phase (between concept and launch) and the commercialization phase (during and after launch) (Lettl 2007; He and Wang 2016; Ahmed et al. 2018).

There are also classifications of innovation by type. Garcia (2014) highlights the following "product/service versus process, radical versus incremental, technological versus administrative, architectural versus modular and disruptive versus sustainable" types while Keeley et al. (2013) listed ten major types of innovation, divided into three groups.

The first group, "configuration", covers the profit model, network, structure and process. The second group, "offering", includes product performance and product system, while the third group, "experience", involves service, channel, brand and customer engagement. Types of innovation differ by field of interest and industry, but most studies (Keeley et al. 2013) have primarily focused on the technology sector, particularly IT, where innovation is intrinsic to products and services (Santoro et al. 2018).

SME innovation processes differ greatly from those of large companies (Bresciani and Ferraris 2014) and enable faster processes, for example in decision-making, in addition to the scope for greater flexibility and less formality. However, there are lower levels of research and development resources available (Van de Vrande et al. 2009; Lee et al. 2010). Thus, Lehtimäki (1991) points out how the aptitude of SMEs for innovation interlinks with the capacities and talents of their senior management. In summary, innovation emerges as essential for the survival and success of companies experiencing fierce competition (Chang et al. 2012).

Huang et al. (2010) posit that open innovation leads to business growth by permitting organizations to leverage more ideas from a variety of external sources.

Therefore, the open innovation model leads to a systematic orientation to openness, in terms of relying on both internal and external resources and exploiting internal and external paths to markets (Chesbrough 2003b; Chaston and Scott 2012).

An important factor spurring the adoption of open innovation stems from the rising costs of technology in many industries (Chesbrough 2003b; Wayne Gould 2012). Furthermore, the open view displays greater alignment with the new "landscape of abundant knowledge" (Chesbrough 2003a, p. 37) and correspondingly reflecting a superior strategic approach in keeping with the newly evolving conditions.

### 2.5. Public Knowledge (Open Innovation) and Private Knowledge (Closed Innovation)

Open innovation subdivides into three core processes: outside-in, inside-out and coupled. This classification provides guidance on how to complement and extend internal innovation processes by establishing an external periphery (Gassmann and Enkel 2004).

In brief, applications of open innovation take place in a range of different ways (Christensen et al. 2005; Durst et al. 2018; Friesike et al. 2015). According to many authors, open innovation approaches are far more effective than the closed innovation alternatives (Rosa et al. 2020; Grama-Vigouroux et al. 2019). In fact, open innovation is not only important to SMEs but also for larger scale companies (Van de Vrande et al. 2009) even while holding greater relevance to the former (Ahmed et al. 2018).

The underlying assumption of the closed innovation model states that successful innovation requires control (Herzog 2008).

In contrast, closed innovation sees firms generate and commercialize their own inventions internally, wherefore the main purpose of intellectual property rights in this context is to protect this knowledge and exclude others (Chesbrough 2003a; Yun et al. 2016).

In this context, closed innovation approaches remain cut off from the external ideas and knowledge that activate direct learning to expand proven knowledge (Kodama and Shibata 2015) Therefore, this knowledge becomes more efficient when taking decisions and actions in accordance with established standards. Thus, the characterization of closed innovation features the efficient implementation of decision-making processes and actions already established, according to proven knowledge (Rajala et al. 2012).

In the closed innovation model, both internal teams and well-defined and intentionally formed inter-organizational networks (Gadde and Mattsson 1987) dominate innovation development. Chesbrough (2003a, 2003b) describes the transition from the closed to the open innovation model, a period of abandoning rigorous internal controls to bring about the significant integration of internal and external components.

### 2.6. The Spinner Innovation Model

According to Figueiredo and Ferreira (2020), the Spinner innovation model focuses on the relationships between knowledge creation, knowledge transfer and innovation, based on the internal and external contexts of SMEs. This model advocates interrelating internal and external factors to develop knowledge intensive solutions (KISs) (Figueiredo et al. 2021). This also remains important whether in scenarios of "open and closed innovation" or "public and private knowledge" and, more particularly, in the context of predictive models of SME innovation (Ibarra et al. 2020; Müller 2019; Rosenbusch et al. 2011; Zeng et al. 2010).

Private knowledge focuses on the internal context while public knowledge encapsulates the external context to bringing about innovation. The Spinner innovation model stems from strategic processes of open innovation through the ongoing relationships of several organizations, including startups, SMEs, universities, knowledge-intensive business services (KIBSs), research centers, government agencies and organizational stakeholders (Figueiredo and Ferreira 2020). This model thereby incorporates the perspective that the available resources and business internationalization theories influence the context of SME international development (Figueiredo et al. 2020).

These relational processes between internal and external variables support SMEs through flows of knowledge, namely the Spinner flows (sectors, SMEs and business) (Figueiredo and Bahli 2021). These flows help SMEs structure the innovation process through striving to develop innovative solutions out of the interactions ongoing among several organizations (Veiga et al. 2021). These flows draw support from established practices, technologies (digitalization, big data, Internet of Things, technological models, data visualization, data guidance, strategic learning, integration of actors, scientific research, diversified environments) and the indicators identifying the steps in knowledge creation, knowledge transfer and knowledge-intensive solutions (Figueiredo et al. 2022). All of these processes derive from the following three drivers: results oriented, entrepreneurial mindset and business transformation (Figure 1).

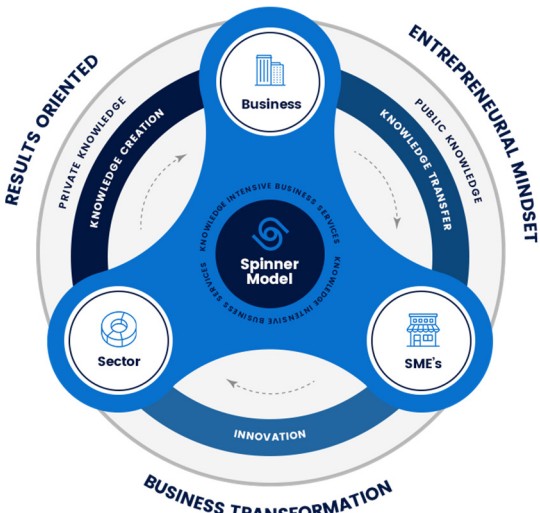

**Figure 1.** The Spinner model, Figueiredo and Ferreira (Figueiredo et al. 2022).

## 3. Method and Results

### 3.1. Method

Taking a sample of small and medium enterprises (SMEs) in the city of Oporto, Portugal, this research study applied the cross industry standard process for data mining (CRISP-DM in short), considered as the most common data mining method (Plotnikova et al. 2022).

The CRISP-DM method represents the main potential alternative proposed in the literature on data mining processes and consists of six major phases: **business understanding, data understanding, data preparation, modelling, evaluation and deployment** (Ghanadbashi et al. 2013).

The business understanding phase involved converting the knowledge into a data extraction problem definition, while the data understanding phase focused on building a table from the sample data in the data warehouse. Data preparation includes all of the activities necessary to establish the final data set to support the raw data modelling process. The development of the data model was performed in accordance with IBM's SPSS Statistics software package, including the "modelling" process, and the automatic discovery of the data combination that reliably predicts the desired result. In the evaluation phase, the model was built, and all phases were reviewed before proceeding to the final

implementation. Finally, we concluded that the implementation phase described all of the results of the model in a way that ensured its reproducibility through future studies.

We collected our data through an online questionnaire developed for the Spinner innovation model (Figueiredo and Bahli 2021). This model applies a 49-factor scale to predict innovation (in SMEs) (Figueiredo and Ferreira 2020; Alawamleh et al. 2018) with the questionnaire spanning two sections: (1) the demographic characteristics of respondents (e.g., academic qualification, gender, marital status and position) and (2) the variables and factors supported by a seven-point Likert scale (1 = strongly disagree and 7 = strongly agree).

The Spinner innovation model used five variables to compose the factors. The variables are (1) private knowledge management (PVKM); (2) public knowledge management (PBKM); (3) knowledge transfer (KT); (4) knowledge creation (KC) and (5) innovation (INN). The (PVKM); (PBKM); (KT) and (KC) are independent variables (X) and (INN) is the dependent variable (Y).

*3.2. Results*

In the initial phase, **"business understanding"**, the main study objective involves converting the knowledge into a data mining problem definition. In our study, the key issue stems from answering the following research question:

RQ: How to predict the innovation of SMEs?

The second phase focused on building up a data table out of the sample data in the data warehouse, i.e., "**data understanding**". The total number of respondents in Oporto was 240 employees at different levels of their careers in SMEs. The representative mining sample reduced the processed total to 208 as some respondent data was lacking (Table 1). This represents an important phase when the researcher becomes familiar with the data and identifies the core issues and establishes the study hypothesis:

**Table 1.** Sample.

| Academic Degree | | |
|---|---|---|
| | N | % |
| High school | 121 | 58.2% |
| Higher education | 87 | 41.8% |
| **Gender** | | |
| | N | % |
| F | 86 | 41.3% |
| M | 122 | 58.7% |
| **Marital Status** | | |
| | N | % |
| Married | 110 | 52.9% |
| Single | 95 | 45.7% |
| Widow | 3 | 1.4% |
| **Position** | | |
| | N | % |
| Assistant | 23 | 11.1% |
| Commercial | 1 | 0.5% |
| Coordinator | 1 | 0.5% |
| Leadership | 2 | 1.0% |
| Management | 127 | 61.1% |
| Operational | 52 | 25.0% |
| Owner | 1 | 0.5% |
| Specialist | 1 | 0.5% |
| N = 208 | | |

H: The Spinner innovation model positively influences the contributions of innovation to SMEs.

Following the method of sampling, we numerically explored the data and identified groupings and refined the discovery process. We then applied the factor analysis technique and reduced the number of factors from 49 to 26 and detected the structure underpinning the relationships among these factors (Walumbwa et al. 2008; Lance et al. 2006). Evaluating the model by the Kaiser–Meyer–Olkin (KMO) test, it returned a result of 0.955 while Bartlett's Test attained a significance at 0.000, as set out in Table 2.

**Table 2.** Exploratory factor analysis (EFA).

| KMO and Bartlett's Test | | |
|---|---|---|
| Kaiser–Meyer–Olkin Measure of Sampling Adequacy. | | 0.955 |
| Bartlett's Test of Sphericity | Approx. Chi-Square | 11,927.466 |
| | df | 1176 |
| | Sig. | 0.000 |

The commonality analysis result was >0.5, as detailed in Table 3 and it obtained an appropriate range in the variable.

**Table 3.** Commonalities.

| | Initial | Extraction |
|---|---|---|
| pvkm_1 | 1.000 | 0.588 |
| pvkm _2 | 1.000 | 0.758 |
| pvkm _3 | 1.000 | 0.755 |
| pvkm _4 | 1.000 | 0.612 |
| pbkm _1 | 1.000 | 0.794 |
| pbkm _2 | 1.000 | 0.784 |
| pbkm _3 | 1.000 | 0.811 |
| pbkm _4 | 1.000 | 0.745 |
| kt_1 | 1.000 | 0.823 |
| kt _2 | 1.000 | 0.810 |
| kt _3 | 1.000 | 0.816 |
| kt _4 | 1.000 | 0.834 |
| kt _5 | 1.000 | 0.821 |
| kt _6 | 1.000 | 0.771 |
| kt _7 | 1.000 | 0.681 |
| kt _8 | 1.000 | 0.823 |
| kt _9 | 1.000 | 0.846 |
| kt _10 | 1.000 | 0.800 |
| kt _11 | 1.000 | 0.680 |
| kt _12 | 1.000 | 0.631 |
| kt _13 | 1.000 | 0.751 |
| kc_1 | 1.000 | 0.709 |
| kc_2 | 1.000 | 0.775 |
| kc_3 | 1.000 | 0.632 |
| kc_4 | 1.000 | 0.769 |
| kc_5 | 1.000 | 0.825 |
| kc_6 | 1.000 | 0.892 |
| kc_7 | 1.000 | 0.819 |
| kc_8 | 1.000 | 0.862 |
| kc_9 | 1.000 | 0.729 |
| kc_10 | 1.000 | 0.745 |

**Table 3.** *Cont.*

|  | Initial | Extraction |
|---|---|---|
| kc_11 | 1.000 | 0.851 |
| kc_12 | 1.000 | 0.888 |
| kc_13 | 1.000 | 0.835 |
| kc_14 | 1.000 | 0.810 |
| kc_15 | 1.000 | 0.827 |
| kc_16 | 1.000 | 0.760 |
| kc_17 | 1.000 | 0.638 |
| inn_1 | 1.000 | 0.746 |
| inn_2 | 1.000 | 0.776 |
| inn_3 | 1.000 | 0.785 |
| inn_4 | 1.000 | 0.830 |
| inn_5 | 1.000 | 0.820 |
| inn_6 | 1.000 | 0.790 |
| inn_7 | 1.000 | 0.759 |
| inn_8 | 1.000 | 0.760 |
| inn_9 | 1.000 | 0.797 |
| inn_10 | 1.000 | 0.769 |
| inn_11 | 1.000 | 0.827 |

Note: Extraction method: principal component analysis.

In terms of variance, the model extracted seven factors with a cumulative variance of 77.462. We followed up this factor analysis by measuring the relationship between the variables according to correlation analysis (Syazali et al. 2019; Belekoukias et al. 2014), (Table 4). The resulting correlation coefficient (significant at the 0.01 level) demonstrates whether changes to one variable result in changes in the other.

**Table 4.** Correlation analysis.

| Correlations | | pvkm | pbkm | kt | kc | inn |
|---|---|---|---|---|---|---|
| pvkm | Pearson Correlation | 1 | 0.415 ** | 0.505 ** | 0.515 ** | 0.433 ** |
|  | Sig. (2-tailed) |  | <0.001 | <0.001 | <0.001 | <0.001 |
|  | N | 208 | 208 | 208 | 208 | 208 |
| pbkm | Pearson Correlation | 0.415 ** | 1 | 0.481 ** | 0.477 ** | 0.666 ** |
|  | Sig. (2-tailed) | <0.001 |  | <0.001 | <0.001 | <0.001 |
|  | N | 208 | 208 | 208 | 208 | 208 |
| kt | Pearson Correlation | 0.505 ** | 0.481 ** | 1 | 0.630 ** | 0.476 ** |
|  | Sig. (2-tailed) | <0.001 | <0.001 |  | <0.001 | <0.001 |
|  | N | 208 | 208 | 208 | 208 | 208 |
| kc | Pearson Correlation | 0.515 ** | 0.477 ** | 0.630 ** | 1 | 0.625 ** |
|  | Sig. (2-tailed) | <0.001 | <0.001 | <0.001 |  | <0.001 |
|  | N | 208 | 208 | 208 | 208 | 208 |
| inn | Pearson Correlation | 0.433 ** | 0.666 ** | 0.476 ** | 0.625 ** | 1 |
|  | Sig. (2-tailed) | <0.001 | <0.001 | <0.001 | <0.001 |  |
|  | N | 208 | 208 | 208 | 208 | 208 |

Note: ** means correlation significant at the 0.01 level (two-tailed).

Subsequently, we analysed the reliability (Figueiredo and Ferreira 2020) of the items applied by the Spinner innovation model scale. Table 5 sets out Cronbach's Alpha result (0.965).

**Table 5.** Reliability statistics.

| **Item-Total Statistics: N of Items 26** | | | | | |
|---|---|---|---|---|---|
| Cronbach's Alpha: 0.963 | | | | | |
| Cronbach's Alpha Based on Standardized Items: 0.965 | | | | | |
| | Scale Mean if Item Deleted | Scale Variance if Item Deleted | Corrected Item-Total Correlation | Squared Multiple Correlation | Cronbach's Alpha if Item Deleted |
| pvkm_1 | 127.77 | 811.200 | 0.505 | 0.497 | 0.964 |
| pvkm_2 | 127.30 | 820.048 | 0.540 | 0.577 | 0.963 |
| pvkm _3 | 127.41 | 819.721 | 0.525 | 0.589 | 0.963 |
| pvkm _4 | 126.69 | 814.284 | 0.644 | 0.573 | 0.962 |
| pbkm _3 | 126.49 | 816.000 | 0.646 | 0.728 | 0.962 |
| pbkm _4 | 126.51 | 816.183 | 0.669 | 0.738 | 0.962 |
| kt_7 | 126.80 | 824.964 | 0.551 | 0.563 | 0.963 |
| kt _8 | 127.10 | 808.236 | 0.667 | 0.820 | 0.962 |
| kt _9 | 127.16 | 805.100 | 0.694 | 0.882 | 0.962 |
| kt _10 | 127.33 | 813.102 | 0.638 | 0.783 | 0.962 |
| kt _11 | 126.71 | 815.831 | 0.703 | 0.630 | 0.962 |
| kc_9 | 126.89 | 804.340 | 0.752 | 0.699 | 0.961 |
| kc _10 | 126.92 | 806.114 | 0.710 | 0.753 | 0.962 |
| kc _11 | 126.88 | 801.202 | 0.774 | 0.871 | 0.961 |
| kc _12 | 126.91 | 800.166 | 0.790 | 0.899 | 0.961 |
| kc _13 | 127.04 | 803.564 | 0.779 | 0.854 | 0.961 |
| kc _14 | 127.09 | 795.127 | 0.796 | 0.803 | 0.961 |
| inn_1 | 126.76 | 808.328 | 0.753 | 0.701 | 0.961 |
| inn_2 | 126.35 | 814.817 | 0.744 | 0.794 | 0.961 |
| inn_3 | 126.63 | 808.709 | 0.755 | 0.785 | 0.961 |
| inn_4 | 126.29 | 812.363 | 0.772 | 0.850 | 0.961 |
| inn_5 | 126.29 | 814.535 | 0.754 | 0.858 | 0.961 |
| inn_6 | 126.46 | 810.771 | 0.763 | 0.832 | 0.961 |
| inn_7 | 126.44 | 818.334 | 0.709 | 0.792 | 0.962 |
| inn_8 | 126.75 | 801.089 | 0.823 | 0.804 | 0.961 |
| inn_11 | 126.64 | 802.781 | 0.801 | 0.808 | 0.961 |

The next phase, "**data preparation**", includes all of the activities necessary to establish the final dataset to support the raw data modelling process. Data preparation involved several facets, including cleaning and transforming the data into a modelling tool. To this end, we selected one independent variable, "innovation", to focus the model selection process on a particular "predictive" direction.

Following preparation, we began the "**modelling**" process by developing the data model in accordance with the IBM SPSS Statistics software package and automatically searched for the data combination that reliably predicted the desired outcome. This technique applies a regression analysis (Faul et al. 2009) to one variable, with the dependent variable (innovation "inn") predicted from the other independent variables (private knowledge management "pvkm"; public knowledge management "pbkm"; knowledge transfer "kt" and knowledge creation "kc"). The application of the ordinary least squares (OLS) test verified the influence of the "pvkm", "pbkm", "kt" and "kc" on "inn", according to the regression model:

$$Y = \hat{\beta}o + \hat{\beta}1 \, \text{pvkm} + \hat{\beta}2 \, \text{pbkm} + \hat{\beta}3 \, \text{kt} + \hat{\beta}4 \, \text{kc}$$

The correlation between the model's independent and dependent variables was strong, with R = 0.753 and it explained 56% of the data variance with R2 = 0.567. This model also increased the prediction of innovation by 56% with R2 change = 0.567 (see Table 6). In terms of the independent residual, the Durbin–Watson test of the model obtained a result of 1.766, which was deemed to be good as it was close to 2.

**Table 6.** Model summary.

| Model Summary [b] | | | | | | | | | | |
|---|---|---|---|---|---|---|---|---|---|---|
| Model | R | R Square | Adjusted R Square | Std. Error of the Estimate | R Square Change | F Change | dif1 | dif2 | Sig. F Change | Durbin–Watson |
| 1 | 0.753 [a] | 0.567 | 0.559 | 0.869 | 0.567 | 66.506 | 4 | 203 | <0.001 | 1.766 |

Note: a. predictors: (constant), kc, pbkm, pvkm, kt; b. dependent variable: inn.

In terms of the model's predictors, Table 7 sets out the ANOVA test results, significant at Sig. < 0.001 ($p < 0.005$).

**Table 7.** ANOVA.

| ANOVA [a] | | | | | | |
|---|---|---|---|---|---|---|
| | Model | Sum of Squares | df | Mean Square | F | Sig. |
| 1 | Regression | 200.784 | 4 | 50.196 | 66.506 | <0.001 [b] |
| | Residual | 153.216 | 203 | 0.755 | | |
| | Total | 354.000 | 207 | | | |

Note: a. dependent variable: inn; b. predictors: (constant), kc, pbkm, pvkm, kt.

Regarding multicollinearity, the relevance of the model's factors, the test reported its non-existence in keeping with tolerance >0.1 and VIF < 10, as Table 8 details.

**Table 8.** Multicollinearity.

| Collinearity [a] | | | |
|---|---|---|---|
| | Model | Collinearity Tolerance | Statistics VIF |
| 1 | (Constant) | | |
| | pvkm | 0.662 | 1.510 |
| | pbkm | 0.699 | 1.431 |
| | kt | 0.533 | 1.876 |
| | kc | 0.528 | 1.892 |

Note: a. dependent variable: inn.

In terms of the predicted value and the standard value, the residuals ranged between −4 + 4 according to the test applied to report on the model outliers, as shown in Table 9.

**Table 9.** Outliers.

| Residual Statistics [a] | | | | | |
|---|---|---|---|---|---|
| | Minimum | Maximum | Mean | Std. Deviation | N |
| Predicted Value | 2.25 | 6.90 | 5.50 | 0.985 | 208 |
| Residual | −3.415 | 2.801 | 0.000 | 0.860 | 208 |
| Std. Predicted Value | −3.302 | 1.417 | 0.000 | 1.000 | 208 |
| Std. Residual | −3.931 | 3.224 | 0.000 | 0.990 | 208 |

Note: a. dependent variable: inn.

Next, in the **"evaluation"** phase, we advanced with the evaluation of the already built model and reviewed all of the phases before proceeding to final deployment. The objective in this phase involves determining whether or not there is any issue yet to appear or which

requires discussion during this process to ensure the study objectives. To ascertain whether the model attains the quality necessary to move onto the next stage of applying the data mining results, we set aside a portion of the data from the sampling stage to test the work on both the retained sample and the sample deployed to develop the model. The variables thereby applied came out in support of the regression model, classed as a **"significant statistical model"** for predicting innovation of SMEs:

$$[F(4.203) = 66.506; p < 0.001; R^2 = 0.567]$$

Thus, "pbkm" and "kc" predict innovation in SMEs:

$$pvkm (\beta = 0.047; t = 0.831; p < 0.407)$$

$$pbkm (\beta = 0.470; t = 8.511; p < 0.001)$$

$$kt (\beta = -0.018; t = -281; p < 0.779)$$

$$kc (\beta = 0.388; t = 6.102; p < 0.001)$$

Finally, considering **"deployment"** as the final phase in the data mining process, and for researchers interested in the data mining process, we describe all of the model results in a way capable of ensuring their reproduction by future studies.

$$Inn = 1.473 + 0.39 \text{ pvkm} + 0.426 \text{ pbkm} - 0.15 \text{ kt} + 0.324 \text{ kc}$$

## 4. Discussion

In this study, the key issue stemmed from a research question: *How to predict the contributions of innovation to SMEs?* In fact, the predictive models hold relevance in many real-life scenarios (Hyndman and Athanasopoulos 2018) and provide the basis for many decision-making procedures (Shim 2000). Through such studies, we are able to verify some cases deploying predictive models, specifically for SMEs (Figueiredo et al. 2020; Alawamleh et al. 2018; Fan et al. 2017; Genty et al. 2015; Kosmidis and Stavropoulos 2014).

In this research project, we applied the Spinner innovation model and formulated the following hypothesis: the Spinner innovation model positively influences the contributions of innovation to SMEs. The Spinner innovation model and its relationship with SMEs and innovation has already been subject to analysis by previous studies (Alawamleh et al. 2018; Figueiredo et al. 2020). This model is purpose designed for SMEs and fosters the creation of knowledge by engaging in internal and external contexts. In this study, we conclude that the variables of the Spinner innovation model, the public knowledge management and knowledge creation, do influence "innovation", in keeping with how the correlations between the model's independent and dependent variables were robust. Hence, in the "evaluation" phase, we correspondingly concluded that the variables applied by the regression model emerged as a "significant statistical model" for predicting the innovation of SMEs. In fact, this model increased the prediction of innovation by 56%. This is particularly true in the case of public knowledge management and knowledge creation, which predict the innovation of SMEs.

Public (open) innovation subdivides into three core processes: outside-in, inside-out, and coupled. This classification provides guidance on how to complement and extend internal innovation processes through recourse to an external periphery (Gassmann and Enkel 2004). In short, open innovation may serve in a range of different ways (Christensen et al. 2005; Friesike et al. 2015) and this study reports that this also represents a means of predicting the innovation of SMEs. In fact, the literature often identifies knowledge management and innovation as the key drivers for the improvement of organizational performance (Ngoc-Tan and Gregar 2018). There are also studies focusing on the interactions between the open innovation and knowledge management of SMEs. Indeed, the importance of studying open innovation has increased, in keeping with recognition of its ability to foster organizational innovation (Durst et al. 2018). Several authors consider open knowledge to

be central to innovation processes (Rosa et al. 2020; Grama-Vigouroux et al. 2019; Van de Vrande et al. 2009). Additionally, open knowledge has played a major role in SMEs, regarding their innovation and performance (Alawamleh et al. 2018). Simultaneously, there are authors who argue that innovation stems more from open knowledge, as this combines the management of both types of knowledge: open and closed (Grama-Vigouroux et al. 2019).

In the case of knowledge creation, we stated above that the creation of knowledge by organizations has become a critical factor to their success and competitiveness. Knowledge creation involves a continuous process through which one overcomes the individual boundaries and constraints imposed by information and past learning by acquiring new contexts, new views of the world and new knowledge (Saenz et al. 2009). By interacting and sharing tacit and explicit knowledge with others, individuals enhance their capacities to define situations and problems as well as applying their knowledge to act and specifically solve whatever the issue/problem is (Nonaka et al. 2006).

In terms of organizational knowledge creation, this means making available and amplifying the knowledge created by individuals, as well as crystallizing and connecting it with the organization's knowledge system (Nonaka and Takeuchi 1995; Nonaka et al. 2006). Knowledge creation spans the abilities of organizations to develop novel and useful ideas and solutions (Marakas 1999) that involve a continuous learning process, enabling organizations to reach beyond their hitherto acquired knowledge by obtaining inputs that generate new visions of reality (Saenz et al. 2009), thus driving more innovative processes in organizational terms and returning greater synergies between organizational members (Nonaka et al. 2006; Nonaka and Takeuchi 1995). In fact, knowledge creating processes enhance the ability to solve problems (Nonaka et al. 2006) to the extent this enables organizations to become more agile in this matter (Marakas 1999), which nurtures innovation and even new realities (Bhatt 2001).

Furthermore, this establishes a "significant statistical model" for predicting innovation in SMEs, particularly in the case of their public knowledge management and knowledge creation.

As regards the theoretical implications of this study, this research adds new results to previous studies on applying the Spinner innovation model insofar as there remains a scarcity of studies on this topic. In fact, a previous study has already demonstrated how private knowledge holds great relevance to innovation (Figueiredo et al. 2022) while this study identifies public knowledge as also holding great relevance.

In practical terms, this study aims to help companies create and manage their knowledge as an engine for innovation, furthermore, drawing attention to the need to know how to manage external information sources and the appropriate public for interacting with. This study holds even greater relevance in the case of SMEs, as this represents the respective context of application for the Spinner innovation model, a company context displaying greater sensitivity to market fluctuations and that should correspondingly strive to leverage more synergies within and beyond their sector of activity.

## 5. Conclusions

In conclusion, we may state that studying the impact of a model that seeks to predict the contribution of certain factors to company innovation is of great relevance, especially when the context involves the SMEs that constitute such a large proportion of Portuguese companies—and thus of such great importance to the economy—irrespective of their greater exposure and vulnerability to market fluctuations. Therefore, the innovation capacities of these companies require analysis and incentivising so that they become a competitive force that is better able to cope with more hostile scenarios, such as that resulting from the COVID pandemic.

Through this study, we were able to conclude that the Spinner innovation model represents a significant model for predicting the innovation of SMEs, particularly in the case of public knowledge management and knowledge creation. This should encourage this company type to devote greater attention to creating and managing their knowledge, espe-

cially in terms of public knowledge which, in turn, suggests the importance of establishing synergistic relationships with other partners.

Regarding limitations, this study stems from a sample of 208 companies that is not highly representative of the universe of Portuguese SMEs and is also limited from a geographic point of view. This limitation stems from how employees were still very focused on the pandemic and its consequences when the questionnaire was distributed, correspondingly, they were often unavailable to collaborate with academic studies. Thus, the results are not easily generalizable although they may constitute an important reference for companies.

In the future, it would be interesting to apply this model to the same sample but in a post-pandemic context, in order to verify whether differences emerge, as we believe there may be significant differences between both contexts, especially as the pandemic has driven new forms of innovation and knowledge management, especially as regards open innovation (Scotti et al. 2020). It would also be highly relevant to carry out a comparative study with another country, which might return important conclusions about different ways of innovating and creating, sharing and managing knowledge. Finally, undertaking a study based on the sector of activity and the region of company location, as a means of verifying the impact of these variables on the management of company innovation and knowledge, also represents an interesting avenue for future studies.

**Author Contributions:** R.F. worked on the project administration, funding, acquisition, supervision, conceptualization, methodology, software, validation, data curation. C.M. and C.H. worked on the formal analysis, investigation, resources, writing—original draft preparation, writing—review and editing, visualization. All authors have read and agreed to the published version of the manuscript.

**Funding:** This paper is financed by National Funding awarded by the FCT—Portuguese Foundation for Science and Technology to the project «UIDB/04928/2020» and NECE-UBI, R&D unit funded by the FCT—Portuguese Foundation for the Development of Science and Technology, Ministry of Education and Science, University of Beira Interior, Management and Economics Department, Estrada do Sineiro, 6200-209 Covilhã, Portugal.

**Data Availability Statement:** All sources of the data used in this article are listed in the references.

**Conflicts of Interest:** The author declares no conflict of interest.

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
