# Peer review of "How to Predict the Innovation to SMEs? Applying the Data Mining Process to the Spinner Innovation Model"

_socsci, doi:10.3390/socsci12020075_

Round 1
Reviewer 1 Report
the topic discussed is important from the point of view of economic development. the adopted research methodology is interesting. I believe that the study should be developed and continued on the basis of a larger territorial coverage of the surveyed entities.
Author Response
Thank you for the opportunity to improve our work namely "Predicting innovation in SMEs: applying the CRISP data mining modelling process to the Spinner Innovation Model factors". We understand the importance of your feedback for our paper. Future studies will be based on a larger territory of the surveyed entities. We appreciated your suggestion. Thank you.
Reviewer 2 Report
1. I can't see clearly your variables and the answer to your hypothesis. Please revised your paper to show regression analysis and answer your hypothesis.
2. I can't see your hypothesis development and the logical relationship between variables.
3. Please check again your English, I found more than 400 English issues on Grammarly.
4. Please see our yellow mark on your paper, you will find my comment. (open with adobe reader).

Author Response
Review Report 2: Comments and Suggestions for Authors
Thank you for the opportunity to improve our work namely "Predicting innovation in SMEs: applying the CRISP data mining modelling process to the Spinner Innovation Model factors". We understand the importance of your feedback for our paper. The changes requested were analysed and implemented in blue color according to the study proposal. Thanks.
- I can't see clearly your variables and the answer to your hypothesis. Please revised your paper to show regression analysis and answer your hypothesis.
Answer1: We insert the text in section 3.1 describing the variables according to the Spinner Innovation Model. The Spinner Innovation used five variables to compose the model. The variables are (1) Private Knowledge Management (PVKM); (2) Public Knowledge Management (PBKM); (3) Knowledge Transfer (KT); (4) Knowledge Creation (KC) and (5) Innovation (INN). The (PVKM); (PBKM); (KT) and (KC) are independent variables and (INN) is the dependent variable.
In addition, we corrected the Hypothesis: The Spinner Innovation Model positively influence the contributions of innovation to SMEs.
- I can't see your hypothesis development and the logical relationship between variables.
Answer2: We have included the explanation in Answer 1. And we already supported the hypothesis development with the Literature Review: Knowledge Management; Knowledge Creation and Transfer; Innovation; Public Knowledge (Open Innovation) and, Private Knowledge (Closed Innovation) and The Spinner Innovation Model
- Please check again your English, I found more than 400 English issues on Grammarly.
Answer3: We revised the text with a proofreader.
- Please see our yellow mark on your paper, you will find my comment. (open with adobe reader).
Answer4: We have made changes throughout the text according to your considerations. We deleted “Figures/Charts” 2,3 and 4.
Reviewer 3 Report
This article empirically analyses the contribution of innovation to the success of SMEs in Portugal. There are few suggestions:
1. Revisit writing there are unclear sentences and typos
2. Please provide more details of the empirical methodologies used in the article
3. Present your empirical analysis in the better format
Author Response
Thank you for the opportunity to improve our work namely "Predicting innovation in SMEs: applying the CRISP data mining modelling process to the Spinner Innovation Model factors". We understand the importance of your feedback for our paper. The changes requested were analysed and implemented in blue color according to the study proposal. Thanks.
Review Report 3: Comments and Suggestions for Authors
Thank you for the opportunity to improve our work namely "Predicting innovation in SMEs: applying the CRISP data mining modelling process to the Spinner Innovation Model factors". We understand the importance of your feedback for our paper. The changes requested were analysed and implemented in blue color according to the study proposal. Thanks.
- Revisit writing there are unclear sentences and typos
Answer 1: We have revised the text to make the sentences clearer.
- Please provide more details of the empirical methodologies used in the article
- Present your empirical analysis in the better format
Answers 2 and 3: We have separated Section 3. Method and Results into two subsections: 3.1 Method and 3.2 Results. We have included a paragraph summarizing the method. We deleted charts in the end of the test.
“The business understanding phase involved converting the knowledge into a data extraction problem definition, while the data understanding phase focused on building a table from the sample data in the data warehouse. Data preparation includes all the activities necessary to establish the final data set to support the raw data modelling process. The development of the data model was performed in accordance with IBM's SPSS Statistics software package, including the "modelling" process, and the automatic discovery of the data combination that reliably predicts the desired result. In the evaluation phase, the model was built, and all phases were reviewed before proceeding to the final implementation. Finally, we concluded that the implementation phase described all the results of the model in a way that ensured its reproducibility through future studies.”.
Round 2
Reviewer 2 Report
I think the paper was improved.